# Graphene Oxide Nanoparticles Having Long Wavelength Absorbing Chlorins for Highly-Enhanced Photodynamic Therapy with Reduced Dark Toxicity

**DOI:** 10.3390/ijms20184344

**Published:** 2019-09-05

**Authors:** Eun Seon Kang, Tae Heon Lee, Yang Liu, Ki-Ho Han, Woo Kyoung Lee, Il Yoon

**Affiliations:** Center for Nano Manufacturing and Department of Nanoscience and Engineering, Inje University, Gimhae 50834, Korea

**Keywords:** photodynamic therapy, graphene oxide, chlorins, covalent and noncovalent bond, singlet oxygen

## Abstract

The long wavelength absorbing photosensitizer (PS) is important in allowing deeper penetration of near-infrared light into tumor tissue for photodynamic therapy (PDT). A suitable drug delivery vehicle is important to attain a sufficient concentration of PS at the tumor site. Presently, we developed graphene oxide (GO) nanoparticles containing long wavelength absorbing PS in the form of the chlorin derivative purpurin-18-*N*-ethylamine (maximum absorption wavelength [λ_max_] 707 nm). The GO–PS complexes comprised a delivery system in which PS was loaded by covalent and noncovalent bonding on the GO nanosheet. The two GO–PS complexes were fully characterized and compared concerning their synthesis, stability, cell viability, and dark toxicity. The GO–PS complexes produced significantly-enhanced PDT activity based on excellent drug delivery effect of GO compared with PS alone. In addition, the noncovalent GO–PS complex displayed higher photoactivity, corresponding with the pH-induced release of noncovalently-bound PS from the GO complex in the acidic environment of the cells. Furthermore, the noncovalently bound GO‒PS complex had no dark toxicity, as their highly organized structure prevented GO toxicity. We describe an excellent GO complex-based delivery system with significantly enhanced PDT with long wavelength absorbing PS, as well as reduced dark toxicity as a promising cancer treatment.

## 1. Introduction 

Graphene, an sp^2^-bonded two-dimensional carbon nanosheet in the form of a honeycomb lattice, is a nanomaterial that has received much attention for diverse applications, such as nanoelectronics, energy storage and conversion, nanocomposite materials, and biomedicine, on the basis of its extraordinary physicochemical properties [1,2,3,4,5]. Among the graphene derivatives, graphene oxide (GO) has been extensively studied in recent years, especially for drug delivery [6,7,8,9,10], biological imaging [11,12], and photothermal therapy (PTT) [13,14,15,16,17]. Furthermore, GO can be easily modified by biomolecules with abundant functional groups (epoxy, hydroxyl, and carboxylic acid units) on the GO nanosheet, resulting in improved properties, such as stability, solubility, biocompatibility, and targeting ability [18,19,20,21]. In addition, the toxicity of GO strongly depends on several factors, such as concentration, and the size, shape, and type of dispersants, among others [22,23,24,25]. Therefore, surface modification of GO significantly affects the toxicity of GO, and can result in reduced or no toxicity [26,27,28]. 

Photodynamic therapy (PDT), which is a non-invasive and patient-specific cancer treatment, uses a photosensitizer (PS) to absorb light of an appropriate wavelength and transfer photon energy to the surrounding oxygen, which generates highly toxic singlet oxygen (^1^O_2_) [29,30,31,32,33,34,35]. Effective nanocarriers exhibiting high cellular uptake into tumor cells have recently been developed by the authors of this study using gold nanoparticles (sphere and rod types) [31,32] and polyoxometalate complexes [33]. GO has been studied for use as a PDT, as well as a combination PDT/PTT [29,36,37,38]. However, these studies typically used only commercially available PSs that do not absorb at long wavelengths; these include chlorin e6 (Ce6, λ_max_ 660 nm) [29,30], zinc (II) phthalocyanine (λ_max_ 670 nm) [37], porphyrin derivative [38], and 2-(1-hexyloxyethyl)-2-devinyl pyropheophorbide-a (HPPH, Photochlor; λ_max_ 665 nm) [39]. Therefore, the development of an effective GO-based PDT system incorporating a PS with long wavelength absorption remains a great challenge. Furthermore, the toxicity of GO is important as a potential delivery vehicle. 

In this study, we tried to reduce the toxicity of GO in the formation of PS-containing GO complexes. We used a synthetic chlorin derivative (purpurinimide) as a long wavelength-absorbing PS (λ_max_ 707 nm) to permit deeper light penetration at tumor sites. In addition, we fabricated two GO complexes using the same PS by two different synthetic methods: covalent and noncovalent bonding. We compared both GO complexes in terms of their synthesis, stability, cell viability, and dark toxicity. Both GO complexes showed significantly enhanced PDT effects compared to those of free PS (chlorin), which was attributed to the excellent delivery effects of GO. The two GO complexes exhibited comparable phototoxicity, whereas the noncovalent GO complex showed lower dark toxicity than that of the covalent GO complex. To the best of our knowledge, this is the first example of covalent and noncovalent GO–PS complexes that use synthetic chlorin molecules with long wavelength absorption. 

## 2. Results and Discussion

### 2.1. Characterizations of Prepared Photosensitizers and GO–PS Complexes

The synthesis of the purpurinimide derivative as a PS was straightforward (Scheme 1a). Methyl pheophorbide-a (MPa) was obtained by extraction of chlorophyll-a paste followed by column separation. MPa was converted to purpurin-18 (P18), and esterification of P18 afforded purpurin-18 methyl ester (P18ME). Finally, purpurin-18-*N*-ethylamine (P18NEA, PS **1**) was formed after removal of the Boc (*tert*-butyloxycarbonyl) group from Boc-protected purpurinimide (P18NEAB). GO was prepared by a modified Hummer’s method [40], and was conjugated to PS **1** using amide formation with the terminal amine unit of the PS for covalent bonding [36,41,42,43,44] and supramolecular hydrophobic/π-π stacking interactions on the surface of sp^2^-bonded nanocarbons in GO for noncovalent bonding [29,37,39,45,46], resulting in two GO complexes: GO–PS **2** and GO–PS **3**, respectively (Scheme 1b).

All reaction steps were monitored by thin-layer-chromatography (TLC) and UV-VIS absorption spectroscopy. The structures of **1**, **2**, and **3** were characterized by ^1^H-NMR, UV-VIS, fluorescence, FT-IR, Raman spectroscopy, thermogravimetric analysis (TGA), high resolution fast atom bombardment mass spectrometry (HRFABMS) spectrometry, and transmission electron microscopy (TEM) (Figure 1 and Appendix A, see the Appendix A). The ^1^H-NMR spectrum of **1** contained two key triplet signals at 4.56 and 3.30 ppm, corresponding to two aliphatic CH_2_ protons between two nitrogen atoms, and a broad singlet signal at 4.87, corresponding to the NH_2_ terminus in the formation of the purpurinimide (Appendix A). HRFABMS analysis of PS **1** confirmed the formation of PS **1** (calculated [M]^+^ = 621.3189; found = 621.3191, Appendix A). The UV-VIS absorption spectra of **1**–**3** and GO in dimethyl sulfoxide (DMSO) are shown in Figure 1a. The absorption spectra of **2** and **3** included the characteristic absorption features of **1**. The λ_max_ values were progressively red shifted for MPa (668 nm), P18 (700 nm), P18ME (700 nm), and PS **1** (707 nm) (Appendix A). In **2** and **3**, the λ_max_ values of the PS **1** core remained unchanged. The UV-VIS peak at 707 nm was used to confirm the successful binding of PS **1** in GO–PS complexes **2** and **3** (Figure 1b), in which the absorbance (after subtraction of absorbance contributed by GO only) was consistent with the concentration of PS **1** in each sample. The successful functionalization of GO–PS **2** was confirmed by FT-IR spectroscopy (see Figure 1c and Appendix A). The FT-IR spectra (Table 1) of GO included peaks at 1723 cm^−1^ (vibrational C = O stretching frequency of surface carboxylic groups), 1623 cm^−1^ (skeletal vibrations of C = C) and 1400–1000 cm^−1^ (C–O stretching and O–H deformations of the hydroxyl and epoxy groups). Upon successful functionalization of GO with the terminal NH_2_ unit from PS **1**, the amide bond C = O stretching frequency appeared at 1629 cm^−1^ (Appendix A) [47]. The fluorescence emission spectra (excitation wavelength, 530 nm) of **1**–**3** showed good fluorescence activity of the PS in **1**–**3** (Figure 1d), which is very useful for monitoring intracellular uptake of the PS. Fluorescence intensity was lower in **2** and **3** than in **1,** because of a fluorescence quenching effect by GO as an electron acceptor. Notably, the fluorescence intensity of **3** (noncovalently bonded PS, quenching 61%) was decreased compared to **2** (covalently bound, quenching 16%) due to the better π–π stacking interactions of **3**. The fluorescence emission peak (λ_max_) of **2** was blue shifted (739 nm), and the peak of **3** was red shifted (779 nm) from that of **1** (772 nm). Raman spectra of GO, **2**, and **3** revealed changes in the characteristic D (the symmetry *A_1g_* mode) and G (the *E_2g_* mode of sp^2^ carbon atoms) bands at 1343.9 and 1586.6, 1352.5 and 1595.1, and 1352.9 and 1586.9 cm^−1^, respectively (Figure 1e) [47]. Importantly, to determine the exact amount of organic content (PS) in the complexes, TGA (Figure 1f) of GO, **2**, and **3** was performed. The weight loss of GO and **1** at 800 °C in N_2_ was 27.2% and 77.7%, respectively. Complex **2** and **3** had a weight loss of 55.8% and 42.3%, respectively, at 800 °C in N_2_. The GO–PS complexes **2** and **3** were determined to contain approximately 43.4 wt % of GO and 56.6 wt % of PS, and 70.1 wt % of GO and 29.9 wt % of PS, respectively, according to the following equations (x and y are the weight percentage of GO and PS, respectively) [8].
0.272x + 0.777y = 0.558  for  **2**(1)
0.272x + 0.777y = 0.423  for  **3**
x + y = 1

The content was used to calculate the constant concentration of the PS in each sample for the cell studies that included 912 nmol/mg in complex **2** and 482 nmol/mg in complex **3**. Therefore, complex **2** (covalently bound) was determined to contain approximately two times (ratio of 1.89) more PS than complex **3** (noncovalently bound). The TEM of GO, **2** and **3** are shown in Figure 1g. The average size of GO and GO–PS complexes **2** and **3** were 96.1 ± 47.7 nm (107.0 ± 5.0 nm by dynamic light scattering, Appendix A), 104.2 ± 27.1 nm, and 187.2 ± 40.1 nm based on TEM, respectively (Appendix A). 

Importantly, the covalent bonding-assisted reaction of GO–PS complex **2** was carried out under a weak acidic condition (pH 4–6) to prevent noncovalent binding of the PS **1** on the GO nanosheet during the synthesis. However, GO–PS complex **3** was formed at neutral pH to maintain the noncovalently bound PS **1** on the GO nanosheet. Figure 2 shows the results of the release test of PS **1** from both GO–PS complexes **2** and **3** (solid samples in water and methylene chloride (MC) layers), in which PS **1** was easily released from GO–PS complex **3** (noncovalently bound) in the acidic condition (pH 4.2), which resulted in a pink colour of the MC layer (Figure 2b). This observation indicated that PS **1** was successfully noncovalently bound on the GO nanosheet in GO–PS complex **3**. GO–PS complex **2** displayed no release of PS at the same pH because of covalent binding alone, which resulted in a colourless MC layer (Figure 2b), which clearly indicated that complex **2** contained only covalently-bound PS without noncovalently bound PS. In GO–PS complex **3**, PS was released at 43% for 1 h, 58% for 3 h, and 63% for 24 h from the pH 4.2 aqueous layer containing acetone into the MC layer, in which delivery ability of **3** for 3 h corresponded with 92% of 24 h. This observation exhibited that **3** had a fast intracellular PS release effect. The pH-induced release of the noncovalently-bound PS (purpurin-18 methyl ester) from the GO surface has been described previously, where the acidic environment (pH 5.4) easily released noncovalently-bound PS from the GO surface [48]. 

### 2.2. In Vitro Photocytotoxicity and Dark Toxicity 

The successful cellular uptake and localization of **1**–**3** in A549 (human lung adenocarcinoma) cells were confirmed by fluorescence images of the PS in each compound using confocal laser scanning microscopy (CLSM, Figure 3). CLSM images revealed that **2** and **3** delivered markedly more PS into the cells than did **1** (PS alone without GO), which was due to the efficacy of GO as a nanocarrier. Most of the PSs were located in the cell cytoplasm [33]. Interestingly, there was no significant difference in cellular uptake results of complexes **2** and **3**, indicating that the difference in loading methods had negligible effects. The cellular uptake mechanisms of GO-based materials are not yet clearly known. However, recent reports have described several mechanisms, such as endocytosis [47], direct penetration of cell membranes [49], and micropinocytosis [50]. Among them, endocytosis is the most common mechanism for the uptake of GO nanosheet complexes, whereas free PS (as an organic molecule) can be internalized into cells by passive diffusion [51].

The photocytotoxicity and dark toxicity of **1**–**3** and GO for A549 cells were evaluated using the 3-(4,5-dimethylthiazol-2-yl)-2,5-diphenyltetrazolium bromide (MTT) assay at concentration ranges of 1.0–20.0 µM for **1** and 1.4–14.0 µM for **2** and **3**. A549 cells (10 × 10^4^ cells/well) were incubated with the PSs for 24 h and photoirradiated for the photocytotoxicity test using a BioSpec LED with a band-pass filter of 610–710 nm (total light dose 2 J·cm^−2^, irradiation time 15 min; Figure 4). After the cells were incubated for 3, 12, and 24 h, their viability (%) was estimated on the basis of the mitochondrial activity of NADH (reduced form of nicotinamide adenine dinucleotide) dehydrogenase (summarized in Appendix A).

Upon photoirradiation, cell viability decreased for all compounds, consistent with the increased concentration as well as incubation time. PS **1** displayed low photocytotoxicity at 3, 12, and 24 h incubation with half maximal inhibitory concentration (IC_50_) values of 12.61, 5.49, and 1.44 µM, respectively (Figure 4a, Table 2). However, on the basis of the excellent delivery effects of the GO nanocarrier, GO complexes **2** and **3** exhibited significantly-enhanced photocytotoxicity (Figure 4b). To evaluate the IC_50_ values of **2** and **3**, cell viability was determined at a low concentration range (0.14–1.05 µM, Figure 4c). IC_50_ values of **2** and **3** at 3, 12, and 24 h incubation were 0.69 and 0.22, 0.48 and 0.21, and 0.31 and 0.20 µM, respectively. The values exceeded those of PS **1** were 18 and 57 times, 11 and 26 times, and 5 and 7 times, respectively (Table 2). To the best of our knowledge, this is the best result in recent PDT research with such a low light dose (total light dose of 2 J·cm^−2^). Interestingly, the IC_50_ results for **2** and **3** (Table 2) showed that the IC_50_ values of **2** decreased, while **3** remained constant. This suggests a difference in the delivery effect, and therefore the time-dependent intracellular uptake using CLSM at 30 min, 3 h, and 24 h was performed. As shown in Figure 5, **3** had a faster delivery compared with **2** at all times. Therefore, the lower and constant IC_50_ values of **3** may be because of the faster delivery effect of **3** based on rapid intracellular release of PS from the GO complex. 

Complexes **1** and **3** exhibited negligible dark toxicity, whereas **2** had dose-dependent dark toxicity similar to the toxicity of free GO [37]. Notably, complex **3** had less dark toxicity than did **2**. This might have been due to the extensive covering of the GO nanosheet in complex **3** with noncovalently-bound chlorin molecules, which formed a highly organized structure of PS that enabled good contact with the flat GO surface through strong π-π stacking. The resulting hydrophobic interactions mitigated the toxicity exhibited by free GO [37].

To evaluate the effect of GO nanosheets in GO complexes **2** and **3**, we used GO alone (with no PS). GO had concentration-dependent dark toxicity and no phototoxicity effect (Appendix A).

### 2.3. Live/Dead Cell Imaging and Singlet Oxygen Photogeneration 

The cell viability results after photoirradiation were confirmed by fluorescence imaging of live (green)/dead (red) cells stained with calcein acetoxymethyl (AM)/propidium iodide (PI) (Figure 6a). Almost all cells treated with GO complexes **2** and **3** were dead (red colour) compared with the smaller number of dead cells treated with **1**, indicating that the PS loaded on GO nanosheets provided more effective photodynamic activity than that of the PS alone.

To quantify the relative photodynamic effects of all compounds in the absence of tumor cells, ^1^O_2_ photogeneration was measured by 1,3-diphenylisobenzofuran (DPBF) as a selective ^1^O_2_ acceptor (Figure 6b) [43,52]. GO complex **2** exhibited higher ^1^O_2_ photogeneration compared with that of the free PS **1**. Notably, compound **2** was almost comparable to methylene blue (MB), a standard ^1^O_2_ sensitizer. It is interesting that even though the fluorescence intensity of **2** was lower (16% quenching) than **1** (Figure 1d), ^1^O_2_ photogeneration of **2** was higher than **1**. 

Interestingly, **3** displayed markedly decreased (61% quenching) fluorescence intensity compared with **1** (Figure 1d), whereas **3** presented slightly lower (6.03%) ^1^O_2_ photogeneration compared with **1**. Complex **3** showed lower ^1^O_2_ photogeneration compared with **1**, whereas **3** exhibited better PDT activity (Figure 4) based on the delivery effect of GO, as well as the release effect of the PS from **3** in the acidic environment of the cells (Figure 2). GO alone displayed no ^1^O_2_ photogeneration. These results proved that the significantly enhanced photodynamic activity of the complexes was attributable to their higher induction of ^1^O_2_ photogeneration by better intracellular penetration and localization of PS molecules than those of free PS. The complexes achieved this better penetration via an endocytosis approach based on the excellent nanocarrier properties of GO.

## 3. Materials and Methods

### 3.1. Materials

Acetone, methylene chloride (CH_2_Cl_2_, MC), methanol (MeOH), and dimethyl ether were purchased from SK Chemical. Potassium hydroxide (KOH), potassium bromide (KBr), anhydrous sodium sulphate, dimethyl sulfoxide (DMSO), sodium bicarbonate, hydrogen chloride (HCl), and sulfuric acid (H_2_SO_4_) were purchased from Samchun Chemical. *N*-(tert-Butoxycarbonyl)-1,2-diaminoethane(*N*-boc-ethylenediamine), 1,3-diphenylisobenzofuran (DPBF), and methylene blue (MB) were purchased from TCI Chemical. 1-Propanol and pyridine were purchased from Burdick and Jacson Chemical. Trifluoro acetic acid (TFA), 1-ethyl-3-(3-dimethylamino-propyl)carbodiimide (EDC), and *N*-hydroxysuccinimide (NHS) were purchased from Sigma-Aldrich Chemical. RPMI-1640 medium, penicillin-streptomycin, and fetal bovine serum (FBS) were obtained from Gibco Life Technologies, Inc. (Grand Island, NY, USA).

### 3.2. General

All reactions were monitored by thin-layer-chromatography (TLC) using Merck 60 silica gel F254 pre-coated (0.2 mm thickness) glass-backed sheets. Silica gel 60A (230–400 mesh, Merck) was used for column chromatography. UV-VIS absorption spectra were recorded on a SCINCO S-3100 UV-VIS spectrophotometer using a 1 cm quartz cuvette. ^1^H-NMR spectra were obtained using a Varian spectrometer (500 MHz) at the Biohealth Product Research Center of Inje University. High resolution fast atom bombardment mass spectrometry (HRFABMS) analysis was conducted using a Jeol JMS700 high resolution mass spectrometer at the Daegu center of the Korea Basic Science Institute (KBSI), Kyungpook National University, Korea. Transmission electron microscopy (TEM) images were obtained using an H-7600 microscope (Hitachi) at Pusan National University. Fluorescence spectra were obtained using a model F7000 fluorescence spectrophotometer (Hitachi) at Gyeongsang National University. Raman spectra were obtained using a LabRAM HR800UV Raman spectrometer at Gyeongsang National University. Thermogravimetric analysis (TGA) was conducted using a TA Instrument Q600, PH470 at the Pusan center of KBSI.

### 3.3. Synthesis of Photosensitizer

#### 3.3.1. Purpurin-18 (P18)

Methyl pheophorbide-a (MPa) was synthesized from chlorophyll-a paste extracted using acidic methanol. MPa (1 g) was dissolved in pyridine (5 ml) and diethyl ether (400 ml), followed by addition of KOH solution (12 g dissolved in 80 ml of 1-propanol). The resulting solution was aerated and stirred for 3 h. The solution was extracted with water (500 ml) and the pH was adjusted to 2 with aqueous 1 M H_2_SO_4_. The acidic solution was separated with CH_2_Cl_2_/H_2_O and the organic layer was evaporated. The purple residue was purified by column chromatography with an eluent of 5% MeOH/CH_2_Cl_2_. ^1^H-NMR (500 MHz, CDCl_3_): δ 9.30, 9.20, and 8.52 (all s and 1H, 10-H, 5-H and 20-H, respectively), 7.78 (dd, *J* = 17.8, 11.6 Hz, 1H, 3^1^CH = CH_2_), 6.23 (d, *J* = 17.8 Hz, 1H, *trans*-3^2^-H), 6.13 (d, 11.5 Hz, 1H, *cis*-3^2^-H), 5.12 (d, *J* = 8.6 Hz, 1H, 17-H), 4.35 (q, *J* = 7.3 Hz, 1H, 18-H), 3.57 (s, 3H, 12-CH^3^), 3.57(s, 3H, 12-CH^3^), 3.47 (m, 2H, 8^1^-CH_2_), 3.28 (s, 3H, 2-CH^3^), 3.05 (s, 3H, 7-CH^3^), 2.79–2.70 (m, 1H, 17^2^-CH_2_), 2.56–2.45 (m, 1H, 17^1^-CH_2_), 1.70 (d, *J* = 7.3 Hz, 3H, 18-CH_3_), 1.56 (t, *J* = 7.6 Hz, 8^2^-CH_3_), −0.04 and −0.28 (all brs and 1H, NH). 

#### 3.3.2. Pupurin-18 Methyl Ester (P18ME)

P18 (300 mg) was dissolved in CH_2_Cl_2_ and stirred at room temperature for 5 min with diazomethane (5 ml, excess). The mixed solution was evaporated and used in the next step without purification. ^1^H NMR (500 MHz, CDCl_3_): δ 9.63, 9.40, and 8.59 (all s and 1H, 10-H, 5-H and 20-H, respectively), 7.91 (dd, *J* = 17.8, 11.5 Hz, 1H, 3^1^CH = CH_2_), 6.32 (d, *J* = 17.9 Hz, 1H, *trans*-3^2^-H), 6.21 (d, *J* = 11.5 Hz, 1H, *cis*-3^2^-H), 5.21 (d, *J* = 8.7 Hz, 1H, 17-H), 4.41 (q, *J* = 7.5 Hz, 1H, 18-H), 3.82 (s, 3H, 12-CH_3_), 3.67 (q, *J* = 7.9 Hz, 2H, 8^1^-CH_2_), 3.61 (s, 3H, 17^2^-OCH_3_), 3.36 (s, 3H, 2-CH_3_), 3.19 (s, 3H, 2-CH_3_), 2.79–2.70 (m, 1H, 17^2^-CH_2_), 2.54–2.41 (m, 2H, 17^1^-H), 2.07–1.96 (m, 1H, 17^2^-CH_2_), 1.76 (d, *J* = 7.4 Hz, 3H, 18-CH_3_), 1.69 (t, *J* = 7.7 Hz, 3H, 8^2^-CH_3_), 0.28 and −0.03 (all brs, 1H, NH).

#### 3.3.3. Purpurin-18-*N*-Ethylamine, PS **1**

P18ME was dissolved in CH_2_Cl_2_ (5 ml) and *N*-(tert-butoxycarbonyl)-1,2-diaminoethane was added to the solution. The resulting solution was stirred at room temperature for 24 h in the dark. The color changed from purple to green. KOH solution (KOH 30 mg/MeOH 5 ml) was added to the solution until the color changed to purple. The solution was then separated with water to remove KOH and the solvent was evaporated. The mixed residue was dissolved in MC and was poured into TFA and stirred for 30 min. The color changed to green and the residue was poured into 2 M Na_2_CO_3_ and the solvent was evaporated. Column separation using an eluent of 10% MeOH/MC afforded the final compound. ^1^H NMR (500 MHz, CDCl_3_): δ 9.62, 9.37, and 8.57 (all s and 1H, 10-H, 5-H and 20-H), 7.91 (dd, *J* = 17.9, 11.5 Hz, 1H, 3^1^CH = CH_2_), 6.29 (d, *J* = 17.9 Hz, 1H, *trans*-3^2^-H), 6.17 (d, *J* = 11.5 Hz, 1H, *cis*-3^2^-H), 5.34 (d, *J* = 9.1 Hz, 1H, 17-H), 4.87 (brs, 2H, NH_2_), 4.56 (t, *J* = 6.5Hz, 1H, N-CH_2_-CH_2_-NH_2_), 4.35 (q, *J* = 7.3 Hz, 2H, 18-H), 3.82 (s, 3H, 12-CH_3_), 3.65 (q, *J* = 13.3 Hz, 2H, 8^1^-CH_2_), 3.56 (s, 3H, 17^2^-CH_2_), 3.30 (t, *J* = 6.9 Hz, 2H, N-CH_2_-CH_2_-NH_2_), 3.17 (s, 3H, 7-CH_3_), 2.75–2.64 (m, 1H, 17^2^-CH_2_), 2.48–2.33 (m, 2H, 17^1^-CH_2_), 2.03 (d, *J* = 8.8 Hz, 3H, 18-CH_3_), 1.67 (t, *J* = 7.6 Hz, 3H, 8^2^-CH_3_), 0.07 and −0.11 (all brs, 1H, NH). UV-VIS (DMSO) λ_max_ (log ε): 419 (4.08), 511 (2.90), 551 (3.34), 651 (2.96), 707 (3.63) nm. HRFABMS: calculated for C_36_H_41_N_6_O_4_ (MH ^+^) 621.3189; found 621.3191. IR (KBr) 3450, 2960, 2922, 2852, 1734, 1681, 1645, 1601, 1544, 1526, 1453, 1261, 1238, 1094, 1062, 801, 634 cm^−1^.

### 3.4. Synthesis of GO and GO–PS Complexes

#### 3.4.1. Synthesis of GO 

GO was prepared by Hummer’s method [40]. IR (KBr) 3175, 2242, 1961, 1743, 1723, 1698, 1623, 1507, 1393, 1372, 1221, 1050, 846, 586 cm^−1^.

#### 3.4.2. Preparation of GO–PS Complex with Covalent Bond, GO–PS **2** (GO:PS = 1:2)

GO (40 mg) was dissolved in DMSO (20 ml) and was sonicated for 5 min. EDC (84 mg) and NHS (62 mg) were added into the GO solution with sonication (5 min) and adjusted to pH 4–6 using an aqueous solution of 1 M NaOH and 1 M HCl. Thereafter, the GO solution was stirred with P18NEA (80 mg) at room temperature for 24 h (dark condition). After the reaction, the mixed solution was dialyzed using a dialysis membrane with a molecular weight cutoff 10.0 kDa in distilled water in the dark for 3 days to remove unbound PS **1** and DMSO, centrifuged several times with water, and washed with cold MC. Finally, the solution was freeze-dried for 3 days, and a dark-purple powder was obtained. UV-VIS (DMSO) λ_max_ (log ε): 420 (4.06), 510 (3.40), 551 (3.54), 648 (3.36), 707 (3.69) nm. IR (KBr): 3206, 2964, 2924, 2853, 1732, 1677, 1629 (carbonyl group on amide bond), 1528, 1242, 1169, 1071, 791, 588 cm^−1^.

#### 3.4.3. Preparation of GO–PS Complex with Noncovalent Bond, GO–PS **3** (GO:PS = 1:2)

GO (40 mg) was dissolved in DMSO (20 ml) and sonicated for 5 min. Thereafter, PS **1** (80 mg) was added into the GO solution. The mixed solution was filled with deionized water (20 ml) and stirred at room temperature for 24 h in the dark. After the reaction, the mixed solution was dialyzed using dialysis membrane with a molecular weight cutoff 10.0 kDa in distilled water in the dark for 3 days to remove unbound PS **1** and DMSO, centrifuged several times with water, and washed with cold MC. Finally, the solution was freeze-dried for 3 days to obtain a dark-purple powder. UV-VIS (DMSO) λ_max_ (log ε): 419 (4.63), 512 (3.75), 551 (3.99), 650 (3.74), 707 (4.22) nm. IR (KBr): 3340, 2958, 2923, 2853, 1736, 1679, 1644, 1527, 1241, 1063, 792, 586 cm^−1^.

### 3.5. Cell Culture and Photoirradiation

A549 human lung carcinoma cells were obtained from the cell line bank of the Seoul National University Cancer Research Center. The cells were grown in RPMI-1640 (Sigma-Aldrich) containing 10% fetal bovine serum and 1% penicillin at 37 °C in a humidified atmosphere of 5% CO_2_ in air. Phosphate-buffered saline (PBS, Sigma-Aldrich), model CK40-32 PH optical microscope (Olympus), SynergyHT fluorescence multi-detection reader (BioTek), trypsin-ethylenediamine tetraacetic acid (EDTA) solution, and an incubator (37 °C, 5% CO_2_) were used in the procedures. PDT was conducted using a BioSpec LED equipped with a band-pass filter of 640–710 nm. 

### 3.6. MTT Assay and Cell Viability

Cells were placed in wells of a 48-well, flat-bottomed microplate, 200 μL, containing 1 × 10^4^ cells. After 24 h, the medium was removed and the plates were washed two times with PBS. PS **1**, GO–PS **2**, GO–PS **3**, and GO in 200 μl of mixed medium were added to wells. After 24 h, the solutions were removed and the plates were washed three times with PBS, followed by the addition of medium (200 μl). The plates were irradiated (2 J·cm^−2^) for 15 min, followed by the MTT assay to evaluate photodynamic activity. MTT solution was added to each well followed by incubation for 1 h and absorbance at 450 nm was measured using a fluorescence multi-detection reader. Incubation occurred for 3, 12, and 24 h following photoirradiation. Each group consisted of three replicate wells for each experiment.

### 3.7. Cellular Accumulation

A549 cells were spread out on confocal dishes (1 × 10^4^ cells/dish) and incubated for 1 day. After 24 h, the medium was removed and dishes were washed three times with PBS. Furthermore, 1 μM of PS **1**, GO–PS **2**, and GO–PS **3** were added to the dishes. After 24 h, the dishes were washed three times with PBS. Fixation solution was added for 10 min. The fixed cells were washed with PBS and stained with diamidino-2-phenylindole (DAPI) (200 nM) to identify the nucleus. Cellular uptake images were shown using CLSM with an LSM 510 META microscope (Carl Zeiss).

### 3.8. Singlet Oxygen Photogeneration

DPBF was used as a selective singlet oxygen (^1^O_2_) acceptor, being bleached upon reaction with ^1^O_2_. Seven sample solutions of DPBF in DMSO (50 μM) containing DPBF only (50 μM, control sample), DPBF + MB (1 μM), DPBF + PS **1** (1 μM), DPBF + GO–PS **2** (1 μM), DPBF + GO–PS **3** (1 μM), and DPBF + GO (1 μM) were prepared in the dark. All samples were put into wells of a 48-well plate and covered with aluminum foil. The plate was irradiated (2 J·cm^−2^) for 15 min. The absorbance of each sample was measured at 418 nm using a fluorescence multi-detection reader. 

### 3.9. Live and Dead Cell Imaging

A549 cells were spread out on a cell culture slide (2 × 10^4^ cells/well) and were incubated for 1 day. The medium was removed and dishes were washed three times with PBS. Thereafter, 1 μM of PS **1**, GO–PS **2**, and GO–PS **3** were added to the dishes, respectively. After 24 h, the slide was irradiated for 15 min (2 J·cm^−2^) and incubated for 24 h. The slide was stained for 1 h using a Live/Dead Cell staining Kit (EZ-View, BIOMAX) and examined using fluorescence microscopy at an excitation wavelength of 490 nm. 

## 4. Conclusions 

We successfully prepared two complexes comprising GO nanoparticles and either covalently and noncovalently bound PS, which is a synthetic and long wavelength-absorbing chlorin derivative (purpurin-18-*N*-ethylamine, PS **1**). The complexes were fully characterized (Table 3). Cell viability determinations demonstrated that both GO–PS complexes **2** and **3** had significantly enhanced PDT activity compared to that of free PS. This enhancement corresponded to an increase in intracellular accumulation via endocytosis based on the excellent delivery features of GO followed by higher ^1^O_2_ photogeneration after photoirradiation in tumour cells, which was confirmed by CLSM and live/dead cell staining (calcein AM/PI) images. The PDT activity of GO–PS complexes **2** and **3** were similar. However, the noncovalently bound GO–PS complex **3** had lower dark toxicity than did **2**. GO–PS complex **3** displayed lower ^1^O_2_ photogeneration compared with free PS, but had better PDT activity based on the delivery effect of GO, as well as the pH-induced release of the PS from the noncovalent GO–PS complex in the acidic environment in the cells. The results will be useful in developing third-generation PSs with long wavelength absorption as well as low dark toxicity based on the excellent delivery system of GO in PDT for cancer.

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
