# Peer review of "Graphene Oxide Nanoparticles Having Long Wavelength Absorbing Chlorins for Highly-Enhanced Photodynamic Therapy with Reduced Dark Toxicity"

_ijms, 2019, doi:10.3390/ijms20184344_

Round 1
Reviewer 1 Report
The manuscript is presenting interesting data about third generation PS. For the efficient PDT, a long wavelenght PS is necessary together with discrete intracellular localization of the PS.These two main problems in PDT are addressed in this work.
The manuscript deserves publication after some minor improvements related to the TEM data.
TEM images should be improved. GO nanoparticles show a broad variation of size. However, it seems that GO complexes have a reduced size. Why?
Have you any explanation about the peculiar shape of compound 3??
Fig 5 a is not supporting what is described in the text.
Author Response
29 August 20, 2019
Ms. Sherry Duan
Assistant Editor
sherry.duan@mdpi.com
International Journal of Molecular Sciences
Dear Ms. Sherry Duan and Reviewers
We greatly appreciate the editor and all of the reviewers have invested in reviewing our manuscript entitled “Graphene Oxide Nanoparticles Having Long Wavelength Absorbing Chlorins for Highly Enhanced Photodynamic Therapy with Reduced Dark Toxicity” (Manuscript ID: ijms-571159).
We also value the thorough and constructive comments from all of the referees, and have taken every effort to address these comments as listed point-by-point below. We believe that the changes they have suggested considerably strengthen and clarify the manuscript. The changes made during the revision are highlighted in yellow-color.
We made some corrections corresponding with the reviewers comments.
We measured TEM images again and changed representative TEM images for improvement and correction. And we corrected average size of GO.
And we did pH-dependent PS release test for GO–PS complex 3 and added related sentences.
In addition, we did time-dependent intracellular uptake study for different delivery rate corresponding to IC50 values of two GO–PS complexes 2 and 3.
I would like to give our sincere thanks to the editor and all of the referees again for their quite wonderful comments and consideration for the manuscript. Please do not hesitate to contact us if you have any questions.
Yours sincerely,
Prof. Dr. Woo Kyoung Lee and Res. Prof. Dr. Il Yoon
--------------------------------------------------------
REVIEWER 1 REPORT
English language and style
( ) Extensive editing of English language and style required
( ) Moderate English changes required
( ) English language and style are fine/minor spell check required
(x) I don't feel qualified to judge about the English language and style
Yes / Can be improved / Must be improved / Not applicable
Does the introduction provide sufficient background and include all relevant references? (x) ( ) ( ) ( )
Is the research design appropriate? (x) ( ) ( ) ( )
Are the methods adequately described? (x) ( ) ( ) ( )
Are the results clearly presented? (x) ( ) ( ) ( )
Are the conclusions supported by the results? (x) ( ) ( ) ( )
Comments and Suggestions for Authors
The manuscript is presenting interesting data about third generation PS. For the efficient PDT, a long wavelength PS is necessary together with discrete intracellular localization of the PS. These two main problems in PDT are addressed in this work.
The manuscript deserves publication after some minor improvements related to the TEM data.
1) TEM images should be improved. GO nanoparticles show a broad variation of size. However, it seems that GO complexes have a reduced size. Why? Have you any explanation about the peculiar shape of compound 3??
Our response: Thanks so much for your good comment. We agree with that GO complexes should be increased size compared with free GO. We are so sorry for wrong calculation of GO and representative images of 2 and 3 in Figure 1g. We measured TEM images again and carefully checked shape of 2 and 3 in TEM images. Therefore, we corrected average size of GO (96.1±47.7 nm in line 126) for the representative image in Figure 1g and changed representative TEM images of GO, and 2 and 3 in Figure 1g. Also we revised in the Supporting Information (Figures S11–S13).
2) Fig 5 a (currently Figure 6a) is not supporting what is described in the text.
Our response: Thanks so much for your comment. However, we think Figure 5a (currently Figure 6a) is a good evidence for different photoactivity result among compounds 1–3 by fluorescence imaging of live (green)/dead (red) cells stained with calcein AM/propidium iodide (PI). Therefore, we would like to maintain this figure as Figure 5a (currently Figure 6a).
------------------
REVIEWER 2 REPORT
English language and style
( ) Extensive editing of English language and style required
( ) Moderate English changes required
(x) English language and style are fine/minor spell check required
Our response: Thanks so much for the comment. However, we already received English language editing from Editage. We have mentioned this editing in Acknowledgments.
( ) I don't feel qualified to judge about the English language and style
Yes Can be improved Must be improved Not applicable
Does the introduction provide sufficient background and include all relevant references? (x) ( ) ( ) ( )
Is the research design appropriate? (x) ( ) ( ) ( )
Are the methods adequately described? (x) ( ) ( ) ( )
Are the results clearly presented? ( ) (x) ( ) ( )
Are the conclusions supported by the results? () (x) ( ) ( )
Comments and Suggestions for Authors
This manuscript describes two new photodynamic therapy compounds using graphene oxide coupled to long wavelength bio-derived photosensitizers. The introduction of graphene oxide appears to improve the cellular delivery of the photosensitizers, allowing higher photodynamic cell toxicity. Further, the distinct formulation have different dark toxicity, showing the potential of design in increasing phot-specific cellular toxicity.
Overall, the work is well presented and clear, however the presentation would benefit from addressing the following concerns:
1) The similarity of the spectroscopic signals of compound 2 and 3 should be notes and addressed in the text. This does not affect the authors conclusions, but it is not clear that any significant distinction can been made between the two using spectroscopy.
Our response: Thanks so much for your comment. However, we already have tried fully measured spectroscopic analysis. Therefore, we think current spectroscopic signals for both compounds 2 and 3 are enough. We have shown extended FT-IR spectra of 2 and 3 at 1800~1000 cm–1 (Figure S9 in the Supporting Information), where we clearly showed the amide bond C=O stretching frequency in 2 (covalent bonding) appeared at 1629 cm–1 (summarized in Table 1). In addition, fluorescence emission spectra in Figure 1d clearly display quite different spectra between 2 and 3. And Raman spectra also present difference between 2 and 3 in Figure 1e.
2) The IC50 results for 2 and 3 (Table 2) suggest a difference in the delivery, given that the IC50 for 3 is constant with time, while 2 deceases. Correlating these results with uptake results (similar to figure 3) would provide an explanation for these results, and may show an improved delivery rate for compound 3, which would be an interesting result.
Our response: Thanks so much for your good comment. We agree with the reviewer’s comment, and we did time-dependent intracellular uptake study for 2 and 3 at various times (30 min, 1 h and 24 h). Therefore, we confirmed that 3 has a faster delivery effect compared with 2, which may be because of pH-dependent rapid release of PS. We added Figure 5 for the time-dependent intracellular uptake study with related sentences in lines 190–195:
“Interestingly, the IC50 results for 2 and 3 (Table 2) showing that the IC50 values of 2 decreased, while 3 is constant. This suggests a difference in the delivery effect, therefore time-dependent intracellular uptake using CLSM at 30 min, 3 h and 24 h was performed. As shown in Figure 5, 3 has a faster delivery compared with 2 at all time. Therefore, the lower and constant IC50 values of 3 may be because of the faster delivery effect of 3 based on rapid intracellular release of PS from the GO complex.”
3) The experiment shown in Figure 2b shows release of PS under acidic conditions. In the conclusions the authors propose that the release of PS from 3 in vivo may contribute to the cytotoxicity. This conclusion would be greatly enhanced by quantification of the disassociation of PS from 3 in pseudo-cytoplasmic conditions. How much PS is release, and at what rate. Such quantification are essential for any application of these technologies, and will provide added context to the mechanistic understanding of the photo-toxicity of these and similar compounds.
Our response: Thanks so much for your good comment. We tried PS release test at pH 4.2 and added related sentences in lines 147–150:
“In GO–PS complex 3, PS was released 43% for 1 h, 58% for 3 h, and 63% for 24 h from pH 4.2 aqueous layer contains acetone into MC layer, in which delivery ability of 3 for 3 h is corresponding with 92% of 24 h. This observation exhibited that 3 has a fast intracellular PS release effect.”
4) Both the presentation and the description of figure 5 are somewhat unclear. The distinction between 2, 3, GO(2) and GO(3) in Fiugre 5 is not clear from either the text or the caption. Further, the in text description is difficult to follow, and the numbers cited do not obviously correspond to the figure data. This section should be rewritten for clarity.
Our response: Thanks so much for your good comment. We agree with this comment, and then we maintained GO(2) result as a representative GO for DPBF absorbance decay in Figure 6b and cell viability results in the Supporting Information (Tables S1–S2). Also we corrected related sentences in lines 215–217:
“To evaluate the effect of GO nanosheets in GO complexes 2 and 3, we used GO alone (with no PS). GO had concentration-dependent dark toxicity and no phototoxicity effect (Tables S1‒S2).”

Reviewer 2 Report
This manuscript describes two new photodynamic therapy compounds using graphene oxide coupled to long wavelength bio-derived photosenstizers. The introduciton of graphene oxide appears to improve the cellular delivery of the photosenstizers, allowing higher photodynamic cell toxicity. Further, the distinct formulation have different dark toxicity, showing the potnetial of design in increasing phot-specific cellular toxicity.
Overall, the work is well presented and clear, however the presentation would benefit from addressing the following concerns:
1) The similarity of the spectroscopic signals of compound 2 and 3 should be notes and addressed in the text. This does not affect the authors conclusions, but it is not clear that any significant distinction can been made between the two using spectroscopy.
2) The IC50 results for 2 and 3 (Table 2) suggest a difference in the delivery, given that the IC50 for 3 is constant with time, while 2 deceases. Correlating these results with uptake results (similar to figure 3) would provide an explanation for these results, and may show an improved delivery rate for compound 3, which would be an interesting result.
3) The experiment shown in Figure 2b shows release of PS under acidic conditions. In the conclusions the authors propose that the release of PS from 3 in vivo may contribute to the cyto-toxicity. This conclusion would be greatly enhanced by quantification of the disassociation of PS from 3 in psuedo-cytoplasmic conditions. How much PS is release, and at what rate. Such quantification are essential for any application of these technologies, and will provide added context to the mechanistic understanding of the photo-toxicity of these and similar compounds.
4) Both the presentation and the description of figure 5 are somewhat unclear. The distinction between 2, 3, GO(2) and GO(3) in Fiugre 5 is not clear from either the text or the caption. Further, the in text description is difficult to follow, and the numbers cited do not obviously correspond to the figure data. This section should be rewritten for clarity.
Author Response

(The authors gave the same response as above.)
